# Genome-Wide Identification and Expression Pattern Analysis of the WNK Gene Family in Apple under Abiotic Stress and *Colletotrichum siamense* Infection

**DOI:** 10.3390/ijms25158528

**Published:** 2024-08-05

**Authors:** Ziwen Wei, Zheng-Chao Wu, Jian Zang, Di Zhao, Wei Guo, Hongyan Dai

**Affiliations:** 1College of Horticulture, Shenyang Agricultural University, 120 Dongling Road, Shenyang 110866, China; weizw2001@163.com; 2Analytical and Testing Center, Shenyang Agricultural University, 120 Dongling Road, Shenyang 110866, China; wuzhengchao66@163.com (Z.-C.W.); zangjian@syau.edu.cn (J.Z.); 2015500089@syau.edu.cn (D.Z.)

**Keywords:** genome-wide identification, gene family, apple, with-no-lysine kinase

## Abstract

With-no-lysine kinase (WNK) is a unique serine/threonine kinase family member. WNK differs from other protein kinases by not having a standard lysine in subdomain II of the universally preserved kinase catalytic region. Conversely, the amino acid lysine located in subdomain I plays a crucial role in its phosphorylation. The WNK family has been reported to regulate *Arabidopsis* flowering, circadian rhythm, and abiotic stress. Eighteen members of the *WNK* gene family were discovered in apples in this research, and they were primarily grouped into five categories on the phylogenetic tree. Conserved domains and motifs also confirmed their identity as members of the WNK family. Promoter cis-acting element analysis indicated their potential role in responses to both abiotic stress and phytohormones. Furthermore, qRT-PCR analysis showed that the expression of *MdWNK* family genes was stimulated to different extents by *Colletotrichum siamense*, NaCl, mannitol, ABA, JA, and SA, with *Colletotrichum siamense* being the most prominent stimulant. *MdWNK* family genes were expressed across all apple tissues, with young fruits showing the greatest expression and roots showing the least expression. The research offered detailed insights into the *MdWNK* gene family, serving as a crucial basis for investigating the biological roles of *MdWNK* genes.

## 1. Introduction

Protein kinases are catalysts that move the phosphate group from the third spot on adenosine triphosphate (ATP) to serine, threonine, and tyrosine residues on proteins, leading to the phosphorylation of the proteins. These kinases can function as molecular switches, initiating or modulating signal transduction pathways by adding or removing phosphate groups through phosphorylation or dephosphorylation reactions. In eukaryotic cells, they contribute greatly in controlling signal transduction processes [1]. Protein kinases preserve a highly conserved catalytic domain across eukaryotic organisms, usually made up of 250–300 amino acids and featuring 12 conserved subdomains interspersed with less preserved areas [2].

The with-no-lysine kinase (WNK) gene family encodes a distinct serine/threonine protein kinase. It lacks a highly conserved catalytic lysine (K) residue in its kinase subdomain II, which is the reason it is referred to as “with no lysine kinase” [3]. Other amino acids replace this residue at the site. The lysine residue at this location is preserved in many other protein kinases and is essential for coordinating activities in the ATP active site [4].

It is suggested that the lysine residue in subdomain I of WNK can replace the lysine residue in subdomain II, which is a key site of phosphorylation and conferring the catalytic activity of WNK [5]. Since WNK shares the closest structural similarity with MAPK protein kinase, it has been referred to as MAPK protein kinase up until now. Xu [3] officially named it WNK1 when they cloned a new mammalian serine/threonine protein kinase from the rat brain cDNA library while studying the extracellular signal transduction pathway of MAPK. Soon after, Verissimo [6] discovered that the human body also possesses a class of protein kinases that share the same structural and functional characteristics as the rat WNK1 kinase, and successfully cloned and isolated four *WNK* family genes on human chromosomes.

In plants, an increasing number of WNK families have been reported [7,8]. They have physiological functions in plant growth and development, including controlling when plants flower, aiding in fruit growth and ripening, affecting root formation, and participating in various abiotic stress processes [9]. According to reports, *Arabidopsis thaliana* contains eleven members of the WNK family, with two of them playing functional roles. AtWNK8 has a negative impact on salt and osmotic stress through the modulation of abscisic acid (ABA)-dependent pathways, as shown by Zhang et al. [10], while AtWNK9 boosts ABA signaling pathways to enhance drought resistance in *Arabidopsis*, as demonstrated by Xie et al. [11]. An examination of the whole genome of rice showed that there are nine members of the WNK family that display varying expression patterns when exposed to different abiotic stresses [12]. Peaches, belonging to the dicotyledonous Rosaceae family, have been found to contain eighteen WNK members [13]. Gene expression profiling analyses revealed that certain peach *WNK* genes may participate in peach fruit development and ripening.

In this research, we identified and comprehensively analyzed eighteen members of the WNK family in apple. The analysis included their physicochemical properties, phylogenetic relationships, chromosome distribution, gene structure, conserved domains and motifs, and promoter cis-acting elements. Furthermore, we used GL-3 tissue culture seedlings to analyze the expression patterns of *MdWNK* genes under various conditions, including salt stress, simulated drought, hormone treatments, and infection with *C. siamense*. We also used field-grown GL-3 materials to analyze the tissue-specific expression of *MdWNK* genes. This research offered a thorough examination of the apple *WNK* gene family, which laid the groundwork for further exploring the role of *WNK* genes in combating both biological and abiotic stressors.

## 2. Results

### 2.1. Identification and Characterization of WNK Gene Family in Apple

An analysis of the whole apple genome revealed eighteen *WNK* gene family members. These eighteen members encoded amino acids from 289 to 745, had protein molecular weights (MWs) ranging from 32.92 to 85.13 kDa, and exhibited an isoelectric point (pI) ranging from 4.80 to 8.64. Except for MdWNK11, the pI values of other WNK family proteins were all less than 7, indicating that they may be acidic proteins. Detailed data on the instability index, etc., are available in Table 1.

### 2.2. Phylogenetic Analysis of the WNK Family

A phylogenetic tree was constructed to elucidate the evolutionary relationship between the WNK family in apple and *Arabidopsis*, using the sequences of eighteen MdWNK proteins and eleven AtWNK proteins. The eighteen members of the WNK family were named based on the evolutionary relationship between MdWNK and AtWNK in phylogenetic trees and the naming results available on the NCBI website (Table 1). According to the tree structure and categorization of AtWNK members, the eighteen MdWNK members were classified into five subfamilies, depicted in Figure 1. Group I, II, III, IV, and V were the names given to these five subfamilies. Group I comprised four individuals (MdMdWNK8, MdWNK8A, MdWNK8B, MdWNK10). Group II comprised four individuals (MdWNK4, MdWNK4A, MdWNK5, MdWNK5A). Group III comprised four individuals (MdWNK2, MdWNK2A, MdWNK9, MdWNK9A). Group IV comprised two individuals (MdWNK3, MdWNK3A), while Group V comprised four individuals (MdWNK11, MdWNK11A, MdWNK11B, MdWNK11C). AtWNK6 and AtWNK7 proteins of *Arabidopsis* were not included in any of the five subfamilies.

### 2.3. Structure Prediction of MdWNK Proteins

Predictions were made for the secondary structures of the eighteen MdWNK proteins, as shown in Appendix A. *α*-helix and random coil were identified as the main secondary structures in the eighteen proteins, accounting for 33.33%~39.67% and 31.83%~51.95%, respectively. Additionally, the secondary structure included an extended strand, accounting for 8.76%~13.43%, and β-turn, which accounts for 3.40%~7.80%. It can be observed that the proportion of β-angle was the lowest.

The 3D structures of the MdWNK protein were predicted by homology modeling, as depicted in Appendix A. These proteins can be categorized into five groups based on their phylogenetic relationships. Through 3D model comparison, it could be intuitively observed that some proteins from the same group had the same or similar 3D structures. For example, MdWNK4 and MdWNK4A, MdWNK5, and MdWNK5A in the second group, and MdWNK2 and MdWNK2A in the third group, respectively, have the same 3D structures. Moreover, MdWNK8, MdWNK8A, and MdWNK8B in the first group and MdWNK11A, MdWNK11B, and MdWNK11C in the fifth group have highly similar 3D structures, respectively. The 3D structure prediction results indicated that the proteins in the same group may have similar functions.

### 2.4. Gene Structure, Conserved Domain, and Motif Analysis of MdWNKs

Gene structure analysis helps us further understand the MdWNK family’s evolutionary relationship. The exons, introns, and conserved domains of eighteen *MdWNKs* were analyzed by mapping the gene structure. As depicted in Figure 2a,b, except for MdWNK8 lacking an untranslated region (UTR), the remaining genes contained coding sequences (CDS), UTRs, introns, and protein kinase (PK) domains that were crucial for the protein kinase family. *MdWNKs* contained a range of two to eight exons, with members of the identical subfamily showing resemblances in the quantity of exons and introns. For example, subfamily II members have seven exons and six introns, suggesting a strong evolutionary connection between WNK4/WNK4A and WNK5/WNK5A.

The MdWNK proteins contained ten conserved motifs, among which eighteen WNK proteins sharing conserved motifs 1 to 5 and motif 8, showing similar locations (Figure 2c). Motif 6 and motif 7 were found in all genes except for those in subfamily V. Motif 9 only existed in subfamily II members, while Motif 10 only existed in subfamily III members. The typical sequence of the specific ten conserved motifs can be observed in Figure 2d. In motif 3, subdomain II of the protein kinase lacked the typical lysine (K) and was substituted by asparagine (N) or cysteine (C). In contrast, lysine is present in subdomain I, which aligns with the concept of with no-lysine (K) kinase. The results of amino acid multiple sequence alignment for conserved domains (protein kinase domain) of all *MdWNK* family genes are shown in Appendix A, revealing a more distinct contrast between the lack of lysine in subdomain II and the existence of lysine in subdomain I.

### 2.5. Chromosomal Localization and Synteny Analysis of MdWNKs

The results of chromosomal localization indicated that the eighteen *MdWNKs* were distributed across eleven chromosomes of the apple genome: *MdWNK10* on chr1, *MdWNK4A* and *MdWNK11B* on chr3, *MdWNK5A* and *MdWNK9* on chr4, *MdWNK8B* and *MdWNK3A* on chr6, *MdWNK11A* on chr9, *MdWNK4* and *MdWNK11C* on chr11, *MdWNK5* and *MdWNK9A* on chr12, *MdWNK2* and *MdWNK8* on chr13, *MdWNK8A* and *MdWNK3* on chr14, *MdWNK2A* on chr16, and *MdWNK11* on chr17 (Figure 3).

We performed synteny analysis on eighteen *MdWNK* genes to identify their replication type. According to the results in Figure 4, seventeen of the eighteen *MdWNK* genes were identified as genome-wide duplication (WGD) or segmental duplication. In contrast, only one gene, *MdWNK8*, was identified as dispersed duplication. These findings suggested that WGD or segmental duplication events were the main type of gene replication of the *MdWNKs*. Additionally, we identified eighteen pairs of gene segmental duplication events among these genes. For example, *MdWNK2* and *MdWNK2A* were identified as synteny genes. Furthermore, in order to gain a deeper comprehension of the evolutionary limitations of the *MdWNK* gene family, we computed the ratio of Nonsynonymous (Ka) to synonymous (Ks) ratios for every *MdWNK* gene pair (Appendix A). Thirteen pairs of genes had Ka/Ks ratios below 1, suggesting that these *MdWNK* genes underwent significant purification and selection pressure throughout their evolution. However, the Ks values of five gene pairs could not be calculated, suggesting a potential high sequence divergence between these genes.

### 2.6. Promoter Cis-Acting Element Analysis of MdWNK Genes

The cis-acting element analysis results indicated that all gene promoters included important cis-acting elements (Figure 5). Out of the eighteen *MdWNK* genes, there was more abiotic stress, biotic stress, and phytohormone-responsive-related cis-acting elements (381 and 220, respectively), and fewer cis-acting elements related to growth and development (34). All eighteen *MdWNK* genes had light-responsive elements and anaerobic-responsive elements. ABA-responsive elements were found upstream of all *MdWNK* genes except for *MdWNK2A* and *MdWNK4*. All genes, except for *MdWNK2*, contained MeJA response elements, which were the most abundant except for light-responsive elements. Therefore, there was speculation that certain *MdWNK* genes could play a role in abiotic and biotic stress responses regulated by jasmonic acid (JA). The cis-acting elements of various subfamilies exhibited similarities in type, but varied in quantity.

### 2.7. Expression Analysis of MdWNK Genes under Different Stress Treatments

To further investigate the potential function of *MdWNK* genes, the relative expressions of *MdWNK* genes in apple “GL-3” tissue culture seedlings treated with *C. siamense*, NaCl, mannitol, ABA, JA, and salicylic acid (SA) were examined, respectively (Figure 6). The results showed significant differences in the relative expression of the eighteen *MdWNK* genes under different treatments. All genes showed significant upregulation when treated with *C. siamense*, as shown in Figure 6a, particularly *MdWNK2*, *MdWNK2A*, *MdWNK3*, *MdWNK8*, and *MdWNK11*, and their homologous genes. At 72 h post inoculation, *MdWNK11A* exhibited a relative expression level approximately 65 times greater than the negative control (i.e., untreated plant). These results indicated that all eighteen *MdWNK* genes were induced by *C. siamense*. Under NaCl treatment, the expression levels of *MdWNK3*, *MdWNK3A*, *MdWNK4*, *MdWNK4A*, *MdWNK9*, and *MdWNK11C* significantly increased, indicating their potential involvement in salt stress regulation (Figure 6b). Under mannitol-simulated drought stress, the inductions of *MdWNK3A*, *MdWNK4*, *MdWNK4A*, *MdWNK5*, *MdWNK5A*, and *MdWNK11* were more pronounced, suggesting their potential involvement in drought stress regulation (Figure 6c). Moreover, the examination of *MdWNK* promoters revealed a significant presence of cis-acting elements responsive to phytohormones. Therefore, this study examined the expression patterns of *MdWNKs* in response to ABA, JA, and SA treatments. As shown in Figure 6d, after exposure to ABA treatment, the majority of genes showed increased expression levels at 24 h, with *MdWNK3A*, *MdWNK4*, *MdWNK4A,* and *MdWNK5* being notably stimulated. After JA treatment, the expression of *MdWNK11C* was significantly upregulated, with the relative expression level reaching 92.5 times at 72 h after treatment (Figure 6e). After SA treatment, the expression of *MdWNK3A*, *MdWNK4*, *MdWNK4A*, *MdWNK5*, *MdWNK5A*, *MdWNK9*, and *MdWNK9A* was significantly upregulated (Figure 6f). In conclusion, the expression of *MdWNK* family genes was induced to varying degrees by *C. siamense*, NaCl, mannitol, ABA, MeJA, and SA.

### 2.8. Tissue-Specific Expression Analysis of MdWNK Genes

To investigate the spatial and temporal expression patterns of *MdWNK* family genes, the presence of *MdWNKs* in various tissues and organs was examined. As shown in Figure 7, eighteen *MdWNK* genes were expressed in roots, young fruits, flowers, shoot apices, and young leaves. Young fruits exhibited the highest expression levels among all *MdWNK* genes. In addition, young stems exhibited high expression levels for ten genes, whereas flowers showed high expression levels for eight genes. The roots exhibited the lowest expression levels in almost all *MdWNK* genes. The expression patterns of homologous genes in various tissues are highly similar, for example, *MdWNK2* and *MdWNK2A*, *MdWNK4* and *MdWNK4A*, *MdWNK5* and *MdWNK5A*, *MdWNK11*, *MdWNK11A*, *MdWNK11B*, and *MdWNK11C*.

## 3. Discussion

Phylogenetic trees provide insights into the evolutionary relationships of organisms and are essential for the study of gene families [15]. The *WNK* gene family in apple was thoroughly analyzed for the first time in this research, revealing a total of eighteen genes. A phylogenetic tree analysis of eighteen *MdWNK* genes and eleven *AtWNK* genes showed that the *MdWNK* genes were classified into five subfamilies. In apple, there were homologous genes corresponding to eight subfamilies in *Arabidopsis*. Still, no genes with the same names as *Arabidopsis WNK1*, *WNK6*, and *WNK7* were found (Figure 2). *AtWNK1* and *AtWNK9* exhibited 72% similarity in their amino acid sequences and were grouped together in a subclade with internal node bootstrap values of 99%. Thus, AtWNK1 and AtWNK9 genes may be considered paralogs of each other. Our *MdWNK* genes were more closely related to *AtWNK9*, so we named them *MdWNK9*. *AtWNK6* and *AtWNK7* did not fall into any of the five subfamilies, indicating that they might have been lost during the evolution of apple. The classification of the five subfamilies was confirmed by analyzing the results of apple and examining the exon/intron structure, conserved domains, and motifs (Figure 3). Members in the identical subfamily display comparable gene structures and motif categories, which are strongly preserved and probably perform similar roles in apple. On the other hand, members from different subfamilies display distinct gene structures and motif specificities, suggesting that they may have various biological functions.

Protein kinases have a well-preserved catalytic region, where subdomain I includes the typical sequence G-X-G-X-X-G-X-G-X-V. However, in WNK, the third glycine (G) is substituted by lysine (K), leading to an altered pattern of G-X-G-X-X-K-X-V [16]. Conversely, other amino acids substitute for the lysine in subdomain II, playing a vital role in phosphorylation and aiding in the anchoring and alignment of ATP [17]. These two domain features of the WNK family can be seen in conserved motif analysis (Figure 2d) and multiple sequence alignment results of the MdWNK family (Appendix A). The amino acid homology of eighteen MdWNK domains reached 77.39% (Appendix A), indicating that they all possessed kinase catalytic activity. It is worth noting that MdWNK11 and its homologs in subfamily V exhibit fewer exons/introns and conserved motifs compared to other MdWNK members. Additionally, they have shorter amino acid lengths, ranging from 289 to 350 (Table 1). This discovery aligns with the findings documented by Cao-Pham et al. [18]. The distinct sequences strongly suggest that the molecular regulatory patterns of WNK11 and homologs differ from those of other WNKs [18].

Studying the cis-acting regions in gene promoters is crucial for comprehending gene regulation and forecasting gene functionalities [19]. Previous reports have highlighted the regulatory function of WNK in circadian rhythm and abiotic stress, particularly in *Arabidopsis* and rice [20,21]. In our analysis of the *MdWNK* promoters, we found a significant presence of light-responsive elements in all genes (Figure 5), which suggests their potential role in regulating circadian rhythms. Additionally, we discovered numerous specific regulatory elements related to biological and abiotic stress and phytohormones, with a notable abundance of ABA and JA response elements known to have crucial roles in plant responses to biological and abiotic stress.

Apple trees, being perennial fruit-bearing plants, face a range of abiotic and biotic pressures as they grow and mature [22,23]. Studies have suggested the significant involvement of *WNK* genes in abiotic stress responses, particularly in relation to salt and drought [9,24,25]. AtWNK8 functions as an inhibitor of salt and osmotic stress responses. *Atwnk8* mutants showed increased levels of proline and stronger levels of CAT and POD activity, resulting in improved resistance to salt and osmotic stress [10]. AtWNK9 and AtWNK8 have contrasting impacts on plant reactions to environmental stress. Increased levels of *AtWNK9* led to higher proline accumulation, promoted the activity of various ABA signaling elements (including *ABI1*, *ABI3*, *ERA1*, and *ABF3*), and improved drought tolerance in transgenic *Arabidopsis* [11]. *GmWNK1*, a gene expressed specifically in soybean roots, was discovered to be decreased in response to treatments with ABA, mannitol, PEG, and NaCl [26]. Expressing *GmWNK1* in *Arabidopsis* led to higher levels of ABA, resulting in transgenic plants with improved resistance to NaCl and osmotic stress during the early stages of growth [24]. The overexpression of *OsWNK9* in *Arabidopsis* led to enhanced resistance to salt, drought, and ABA-induced stress [25]. Additionally, transgenic plants showed elevated levels of endogenous ABA and the upregulated expression of genes associated with abiotic stress and ABA signaling when exposed to salt and drought conditions, in contrast to the non-transgenic control. The results indicate that OsWNK9 could control the response to salt stress and drought by using an ABA-dependent mechanism. These results highlight the significant impact of *WNK* genes on plant adaptation to stress. Our research showed that the activation of multiple *MdWNKs* genes occurs in response to different biological pressures and hormone applications, suggesting that *MdWNKs* could be important in responding to abiotic stress factors, supporting earlier research findings [8,27].

Nevertheless, there are few reports on regulating biological stress by WNK proteins. Recently, it has been reported that the expressions of several *GhWNK* genes were induced upon infection with *Verticillium dahliae*, indicating their potential involvement in the immune response of cotton plants to *V. dahliae* [8]. Dunker et al. reported the participation of *AtWNK2* in the plant–fungus interaction process [28]. The expression of the target gene *AtWNK2* was downregulated by the small RNA of pathogenic *Hyaloperonospora arabidopsidis* through RNA-induced silencing. The *Arabidopsis* gene knockout mutant *atwnk2* showed increased susceptibility, indicating that AtWNK2 positively regulates plant immunity. While the function of AtWNK2 is still not understood, and its involvement in other diseases has not been established. Furthermore, there have been reports of an interaction between AtWNK2 and AtWNK11, hinting at their possible participation in a series of kinase reactions, in which one WNK may control the function of the other [29]. The findings from our study indicated that *MdWNK2*, *MdWNK11*, and their homologs were notably activated by *C. siamense* (Figure 6a). This suggests their potential involvement in the regulatory effect of apple on *C. siamense* through interactions and kinase cascade, which requires further exploration for validation.

## 4. Materials and Methods

### 4.1. Identification of WNK Family Members in Apple

In this study, two methods were used to identify the *WNK* family members in apples. Initially, we employed TBtools II v2.030 software to perform a BLASTP inquiry on the local apple database (Version GDDH13 v1.1 from GDR database), utilizing eleven *Arabidopsis* WNK protein sequences (from TAIR database) as query sequences, and setting an e-value threshold of 1 × 10^−10^ [30]. Next, we used the obtained apple WNK sequence as queries to conduct a secondary BLASTP search on the Swiss-Prot database. Due to the absence of Pfam ID for WNK proteins in the Pfam database, we conducted multiple sequence alignments on the *Arabidopsis* WNK family members. Subsequently, we established a hidden Markov model to screen for members of the WNK family in apples. Redundant sequences were then eliminated based on recognition motifs and chromosome positions. In order to confirm the dependability of our findings, we utilized the online platform InterPro (https://www.ebi.ac.uk/interpro/, accessed on 17 January 2024) to validate the potential WNK sequences and ensure the presence and completeness of the WNK domain.

### 4.2. Phylogenetic Analysis

The phylogenetic trees in this study were constructed using MEGA11.0 software, utilizing the maximum likelihood method and JTT+G model with 1000 bootstrap tests [14]. Subsequently, all of the phylogenetic trees were visualized by using the ChiPlot online site (https://www.chiplot.online/, accessed on 23 January 2024).

### 4.3. Sequence Analysis of MdWNK Family Members

The molecular weight, isoelectric point, and other information on MdWNK proteins were calculated using ExPASy online tools (http://web.expasy.org/protparam/, accessed on 18 January 2024). The secondary structures of MdWNK proteins were predicted based on the SOPMA website (https://npsa-prabi.ibcp.fr/cgi-bin/npsa_automat.pl?page=npsa_sopma.html, accessed on 19 January 2024). The 3D structures of MdWNK proteins were established using the Swiss Model online website (https://swissmodel.expasy.org/, accessed on 28 July 2024). The exon–intron structures of *MdWNK* family members were sketched using the Tbtools II v2.030 tool. The conserved motifs of MdWNK proteins were obtained by the MEME 5.0.2 tool. Multiple conserved domain sequences of MdWNK protein were aligned with the MUSCLE Wrapper tool, and similar amino acids were highlighted. The results were visualized using Jalview 9.0.5 software.

### 4.4. Promoter Cis-Acting Element Analysis

The promoter region for each *MdWNK* gene was extracted from the apple genome at a distance of 2000 bp fragments upstream of the translation initiation site. To identify cis-acting elements, MdWNK promoter sequences were supplied to the PlantCARE website [31]. Then, an Excel table was used to create statistics on each MdWNK gene containing various types of cis-acting elements, and Tbtools II v2.030 software was used to draw a heat map of the statistical results.

### 4.5. Chromosomal Localization and Synteny Analysis

The mapping of *MdWNK* genes to the apple chromosomes was carried out using TBtools II v2.030 software and the details were from the apple genome database. Syntenic blocks were identified and distinct duplication events were distinguished using BLASTP and MCScanX module, following the methodology outlined by Wang et al. [32]. The synteny relationships of orthologous *MdWNK* genes were displayed using the Dual Synteny Plotter module [33]. The Ka and Ks values for each duplicated *MdWNK* gene pair were calculated using KaKs_Calculator 2.0 [34].

### 4.6. Plant Material and Stress Treatment

GL-3 apple (*Malus domestica*) materials for tissue-specific gene expression were cultivated at Shenyang Agricultural University under a natural environment. Samples of roots, young leaves, young fruits (approximately 80 days post full bloom), flowers, and shoot apices were collected. The apple materials utilized for stress and hormone treatment were GL-3 tissue culture seedlings that thrived in the growth chamber.

For the hormone and abiotic stress treatment experiment, GL-3 tissue culture seedlings, cultured for about one month, were appropriately pruned and then cultured in MS medium containing 30 μM ABA, 100 μM JA, 100 μM SA, 200 mM NaCl, and 300 mM mannitol. The treated leaves were consecutively gathered at intervals of 0 h, 12 h, 24 h, 48 h, and 72 h.

For the biotic stress treatment experiment, *C. siamense* was grown in PDA medium at 28 °C in the dark for a week. After scraping off the hypha, the *C. siamense* was cultured for an additional week at 30 °C in darkness. The conidia were diluted with sterile water to a concentration of 10^7^~10^8^ conidia mL^−1^ and then sprayed onto the detached apple leaves. The treated leaves were placed under constant temperature and light, and the incidence was observed daily. Specimens were gathered at intervals of 0 h, 24 h, 48 h, and 72 h.

### 4.7. RNA Extraction and Quantitative Real-Time PCR (qRT-PCR)

RNA extraction and reverse transcription were performed as specified by Chang et al. [35]. The reaction system, reagent, and instrument used for qRT-PCR assay were the same as the method of Chen et al. [36]. The experiment utilized primers that were detailed in Appendix A, with *MdActin* (CN938023) chosen as the constitutive control. Gene expression levels were calculated by the 2^−ΔΔCT^ method with each sample undergoing three biological replicates [37]. Statistical analysis was conducted with the DPS 7.05 program.

## 5. Conclusions

This study identified eighteen *MdWNK* family genes in the apple genome. Phylogenetic analysis revealed that the eighteen *MdWNK* genes can be classified into five subfamilies, consistent with the classification of their exon/intron structure, conserved domains, and motifs. An examination of cis-acting elements in promoters indicated that *MdWNKs* could play a significant role in plant responses to different stressors and hormones. Hormone and stress treatment experiments demonstrated that varying degrees of *MdWNK* genes were induced by *C. siamense*, NaCl, mannitol, ABA, JA, and SA. An analysis of gene expression in various apple tissues and organs showed that young fruits had the highest expression levels of all *MdWNK* genes. This research supplied abundant information for understanding *MdWNK* gene functions.

## Figures and Tables

**Figure 1 ijms-25-08528-f001:**
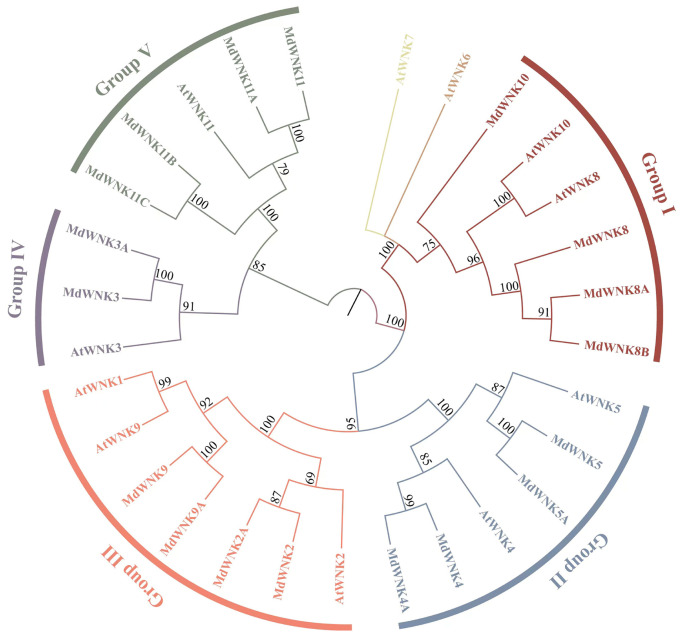
Phylogenetic analysis of WNK proteins from *Arabidopsis* and apple. Different colored branches represent different groups.

**Figure 2 ijms-25-08528-f002:**
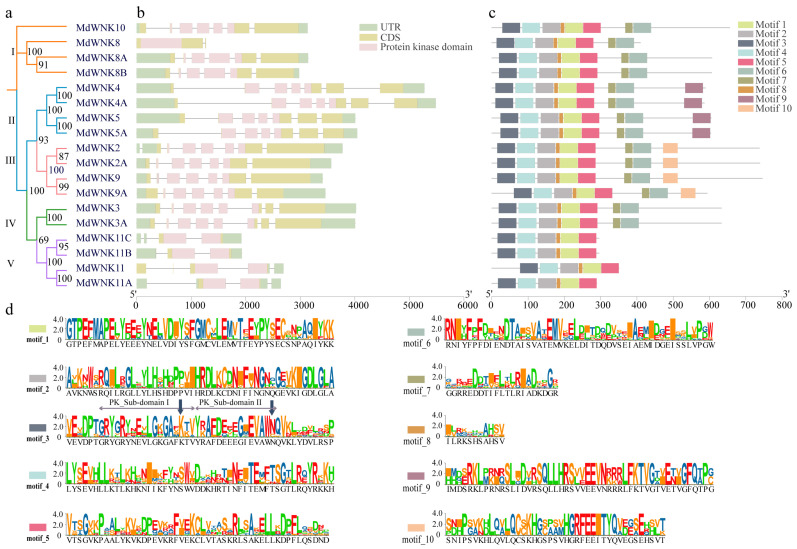
Gene structure and conserved protein motifs of *MdWNKs*. (**a**) Phylogenetic tree of eighteen MdWNK proteins. (**b**) The gene structures of the *MdWNKs*. The green, yellow, and pink boxes represent the UTRs, CDSs, and protein kinase domains, respectively. (**c**) Conserved motif distribution of MdWNK proteins. Different colored boxes indicate different motifs. (**d**) The sequence logos of the ten conserved motifs. The stack height represents each amino acid’s relative frequency at that position. Based on previous research [9,14], subdomains I and II of protein kinases are denoted by the horizontal arrows above, and the positions of lysine (K) in subdomain I and the substituted lysine (K) in subdomain II are marked with arrows.

**Figure 3 ijms-25-08528-f003:**
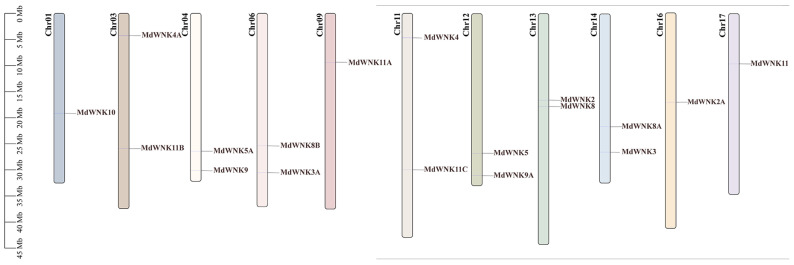
Chromosome distribution of *MdWNKs* on the apple genome.

**Figure 4 ijms-25-08528-f004:**
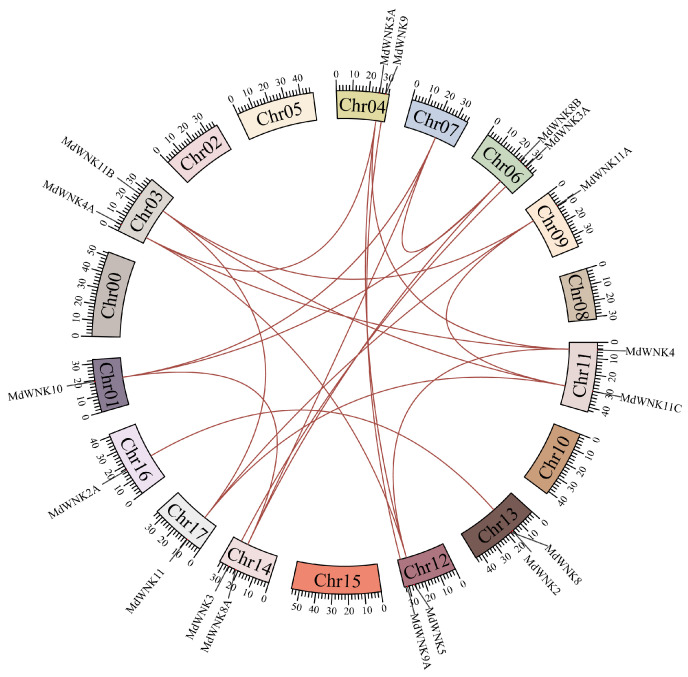
Synteny analysis of *MdWNK* genes. Segmental duplicated gene pairs are represented by brown lines between chromosomes.

**Figure 5 ijms-25-08528-f005:**
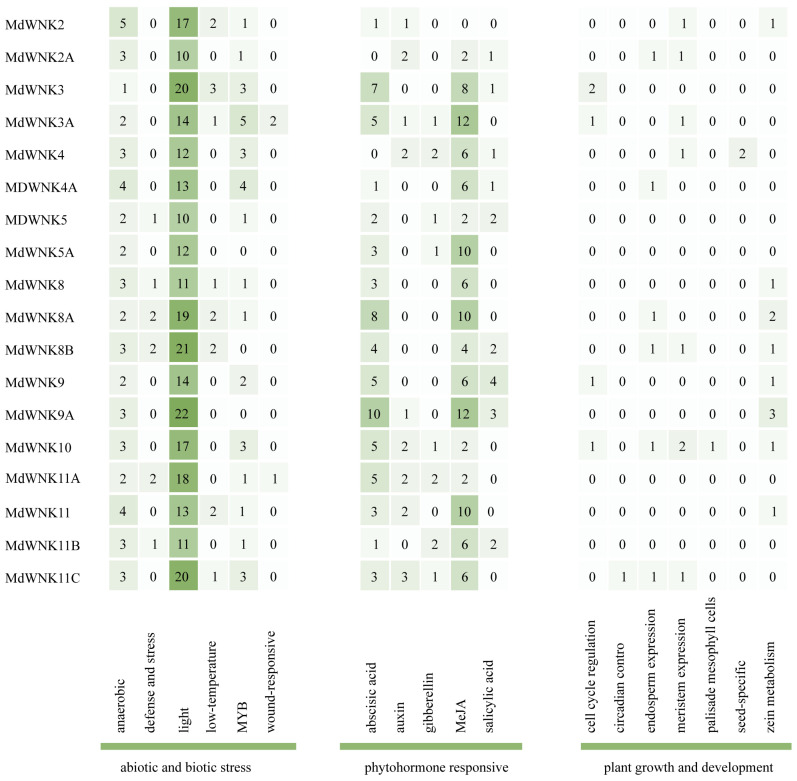
Analysis of the cis-elements in the promoters of the *MdWNK* genes. The numbers represent the number of cis-acting elements detected in the promoter region of each *MdWNK* gene.

**Figure 6 ijms-25-08528-f006:**
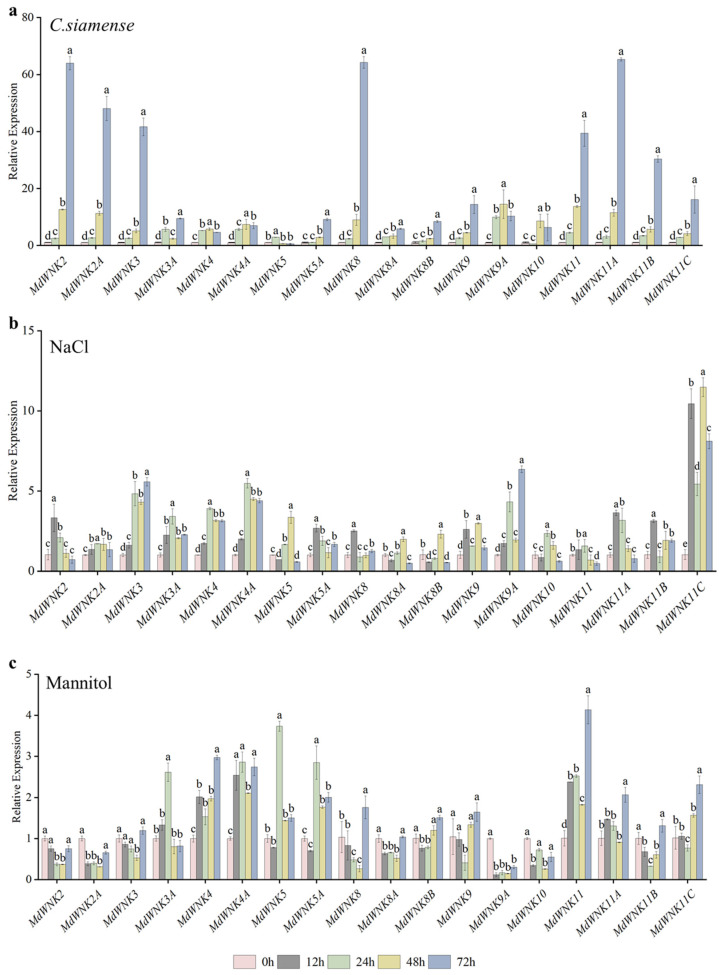
Gene expression patterns of *MdWNKs* under different stress treatments. (**a**–**c**) Relative expression of *MdWNKs* under *C. siamense* inoculation (10^7^~10^8^ conidia mL^−1^), NaCl (200 mM), and mannitol (300 mM). (**d**–**f**) Relative expression of *MdWNKs* under ABA (30 μM), JA treatment (100 μM), and SA treatment (100 μM). Normalization of qRT-PCR data was performed with the *MdActin* gene and presented relative to 0 h (negative control). The results represent the average values with standard deviations, different letters indicate significant differences (n = 3, *p* < 0.05).

**Figure 7 ijms-25-08528-f007:**
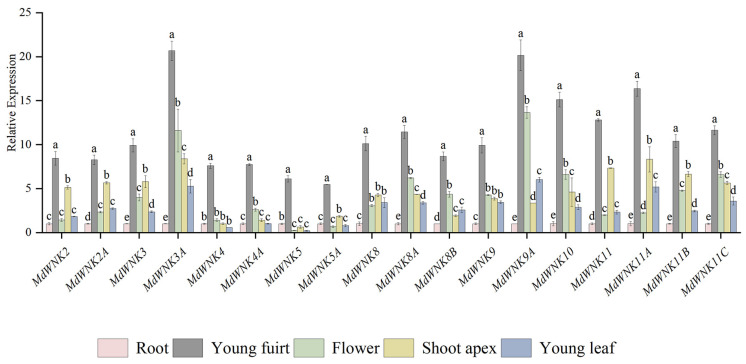
Gene expression patterns of *MdWNKs* in different apple organs. Normalization of qRT-PCR data was performed with the *MdActin* gene and presented in comparison to the root. The results represent the average values with standard deviations, different letters indicate significant differences (n = 3, *p* < 0.05).

**Table 1 ijms-25-08528-t001:** Detailed information about the members of the apple WNK gene family.

Gene Name	Gene ID	Protein Length (aa)	MW (kDa)	pI	Instability Index	Aliphatic Index	Grand Average of Hydropathicity
*MdWNK2*	MD13G1192500	737	84.10	5.35	39.67	76.61	−0.636
*MdWNK2A*	MD16G1193000	738	83.94	5.24	41.25	76.12	−0.604
*MdWNK3*	MD14G1170800	632	70.65	4.87	48.31	79.18	−0.618
*MdWNK3A*	MD06G1165300	632	70.87	4.8	47.1	80.08	−0.633
*MdWNK4*	MD11G1055100	588	66.47	6.22	39	79.59	−0.464
*MdWNK4A*	MD03G1053500	585	66.38	5.99	41.31	80.17	−0.48
*MdWNK5*	MD12G1186000	604	68.82	5.45	51.93	78.11	−0.559
*MdWNK5A*	MD04G1172700	603	68.91	5.86	52.58	74.53	−0.621
*MdWNK8*	MD13G1200200	409	46.34	5.21	34.43	93.37	−0.267
*MdWNK8A*	MD14G1136200	606	67.62	5.14	51.91	80.1	−0.416
*MdWNK8B*	MD06G1114800	605	67.62	5.19	44.31	82.63	−0.44
*MdWNK9*	MD04G1220500	745	85.13	5.08	43.15	77.99	−0.592
*MdWNK9A*	MD12G1236900	593	67.85	5.31	39.43	77.22	−0.475
*MdWNK10*	MD01G1085200	655	73.37	5.49	39.88	84.52	−0.382
*MdWNK11*	MD17G1112400	350	39.96	8.64	43.63	91.63	−0.33
*MdWNK11A*	MD09G1121400	289	32.92	5.89	30.13	87.68	−0.381
*MdWNK11B*	MD03G1189000	295	33.73	5.47	37.06	83.9	−0.465
*MdWNK11C*	MD11G1205200	295	33.85	5.4	37.59	82.54	−0.466

## Data Availability

All data generated or analyzed during this study are included in this article and the Appendix A.

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
