# Peer review of "Genome-Wide Identification and Expression Pattern Analysis of the WNK Gene Family in Apple under Abiotic Stress and Colletotrichum siamense Infection"

_ijms, 2024, doi:10.3390/ijms25158528_

Round 1

Reviewer 1 Report

Comments and Suggestions for Authors

Title: Genome-Wide Identification and Expression Pattern Analysis of the WNK Gene Family in Apple under Abiotic Stress and Colletotrichum siamense Infection

Comments:

The study identified eighteen WNK genes in apples, analyzing their structure, expression patterns, and responses to various stressors. It concluded that these genes are involved in abiotic stress responses and pathogen defense, providing a foundation for further research on their biological roles. It is well-structured and the figures are made professionally. However, there are some small grammar mistakes that need author’ s attention. They are as follows:

1.      P1, L12; ‘With No Lysine kinase (WNK)’ should be ‘The With No Lysine kinase (WNK)’. It applies to the whole manuscript.

2.      P4, Fig. 4; the legend needs more detailed information, such as which method was used for constructing the phylogenetic analysis.

This manuscript presents several analyses of the MdWNK family, providing extensive information on related gene functions. Including phenotypic results from transgenic MdWNK overexpression and RNA silencing lines would strengthen the conclusions.

Comments on the Quality of English Language

The English is professional.

Author Response

The study identified eighteen WNK genes in apples, analyzing their structure, expression patterns, and responses to various stressors. It concluded that these genes are involved in abiotic stress responses and pathogen defense, providing a foundation for further research on their biological roles. It is well-structured and the figures are made professionally. However, there are some small grammar mistakes that need author’ s attention. They are as follows:

comments 1: P1, L12; ‘With No Lysine kinase (WNK)’ should be ‘The With No Lysine kinase (WNK)’. It applies to the whole manuscript.

Response 1: I agree. it has been modified in P1, L12.

comments 1 :P4, Fig. 4; the legend needs more detailed information, such as which method was used for constructing the phylogenetic analysis.

Response 2: Figure 4 showed the Synteny analysis of MdWNK genes. The method of Synteny analysis had been given in section “4.5. Chromosomal Localization and Synteny Analysis”. And the method for constructing Phylogenetic analyses had been given in section “4.2. Phylogenetic Analysis”.

Reviewer 2 Report

Comments and Suggestions for Authors

The authors do a good descriptive job identifying the 18 MdWNK genes from the apple genome. Overall, it is a good piece of work that provides the classic data for this type of study: gene identification, phylogenetic analysis, gene structure and motifs, chromosomal localization... Additionally, they conduct a good transcriptomic study considering various abiotic stresses, tissues, and times, which completes and defines the genes and the study. As I have mentioned, I think it is a good piece of work, with good structure and organization, but I would like some points to be addressed before it can be accepted, all of which are minor but relatively important:

  1. I would like a better description of how the promoter analysis was conducted. The description in materials and methods seems somewhat sparse to me.
  2. I would like the protein structure to be shown at least once. As seen in the domains they name, there are 5 groups; perhaps they could include 5 representative 3D structures of one member from each group.

Author Response

The authors do a good descriptive job identifying the 18 MdWNK genes from the apple genome. Overall, it is a good piece of work that provides the classic data for this type of study: gene identification, phylogenetic analysis, gene structure and motifs, chromosomal localization... Additionally, they conduct a good transcriptomic study considering various abiotic stresses, tissues, and times, which completes and defines the genes and the study. As I have mentioned, I think it is a good piece of work, with good structure and organization, but I would like some points to be addressed before it can be accepted, all of which are minor but relatively important:

comments 1: I would like a better description of how the promoter analysis was conducted. The description in materials and methods seems somewhat sparse to me.

Response 1: We agree. The methodology for promoter analysis has been revised and enhanced, please refer to page 14 the section“4.4. Promoter Cis-Acting Elements Analysis”.

comments 2: I would like the protein structure to be shown at least once. As seen in the domains they name, there are 5 groups; perhaps they could include 5 representative 3D structures of one member from each group.

Response 2: We agree. Thanks for your suggestion. We have added the 3D structural analysis of 18 MdWNK proteins, see Supplementary Figure S1 for details. The results showed that the proteins from the same group had the same or similar 3D structure. Since the grouping of 3D structures is based on evolutionary relationship, I adjusted the secondary Structure and 3D structure prediction of MdWNK to the end of the “2.2. Phylogenetic Analysis of the WNK Family”, as detailed in the section "2.3. Structure prediction of MdWNK proteins" on page 4.